# Role of Extracellular Vesicles in Substance Abuse and HIV-Related Neurological Pathologies

**DOI:** 10.3390/ijms21186765

**Published:** 2020-09-15

**Authors:** Katherine E. Odegaard, Subhash Chand, Sydney Wheeler, Sneham Tiwari, Adrian Flores, Jordan Hernandez, Mason Savine, Austin Gowen, Gurudutt Pendyala, Sowmya V. Yelamanchili

**Affiliations:** Department of Anesthesiology, University of Nebraska Medical Center, Omaha, NE 68198, USA; katherine.odegaard@unmc.edu (K.E.O.); subhash.chand@unmc.edu (S.C.); sydney.wheeler@unmc.edu (S.W.); sam.tiwari@unmc.edu (S.T.); a.flores@unmc.edu (A.F.); jordan.hernandez@unmc.edu (J.H.); mason.savine@unmc.edu (M.S.); austin.gowen@unmc.edu (A.G.); gpendyala@unmc.edu (G.P.)

**Keywords:** extracellular vesicles (EV), drugs of abuse, HIV, methamphetamine (METH), cocaine, nicotine, opioids, alcohol, microRNA (miRNA), CNS disease

## Abstract

Extracellular vesicles (EVs) are a broad, heterogeneous class of membranous lipid-bilayer vesicles that facilitate intercellular communication throughout the body. As important carriers of various types of cargo, including proteins, lipids, DNA fragments, and a variety of small noncoding RNAs, including miRNAs, mRNAs, and siRNAs, EVs may play an important role in the development of addiction and other neurological pathologies, particularly those related to HIV. In this review, we summarize the findings of EV studies in the context of methamphetamine (METH), cocaine, nicotine, opioid, and alcohol use disorders, highlighting important EV cargoes that may contribute to addiction. Additionally, as HIV and substance abuse are often comorbid, we discuss the potential role of EVs in the intersection of substance abuse and HIV. Taken together, the studies presented in this comprehensive review shed light on the potential role of EVs in the exacerbation of substance use and HIV. As a subject of growing interest, EVs may continue to provide information about mechanisms and pathogenesis in substance use disorders and CNS pathologies, perhaps allowing for exploration into potential therapeutic options.

## 1. Introduction

### 1.1. Extracellular Vesicles

Extracellular vesicles (EVs) are a broad, heterogeneous class of membranous lipid-bilayer vesicles that facilitate intercellular communication throughout the body. Secreted from all cell types, these cargo carriers have become important targets of investigation in various fields of study for their potential role in disease pathologies, drug-delivery systems, and therapeutics [1,2]. For the purpose of this review, all three classes of EVs—exosomes (30–150 nm), microvesicles (100–500 nm), and apoptotic bodies (500–5000 nm)—are collectively referred to as EVs, as endorsed by the International Society for Extracellular Vesicles [3]. EVs carry a variety of cargo types, including proteins, lipids, DNA fragments, and a variety of small noncoding RNAs, including miRNAs, mRNAs, and siRNAs [4,5]. The contents of EVs are reflective of the intracellular environments of their host cells, and EVs are released by both healthy and diseased cells [6]. EVs can transfer these cargoes from host cells to recipient cells, inducing functional transformations within recipient cells [7,8,9]. Regulation of EV secretion remains an active area of study, although certain stimuli and cellular conditions have been implicated in triggering EV release from different cell types [10].

EVs play a role in various aspects of healthy physiology, including immune responses [11,12], embryonic stem-cell communication during embryo implantation [13], and exercise [14,15]. EVs also shuttle essential biomolecules between cells that are critical for intercellular communication [16], antigen presentation [17], and signal transduction [18]. Moreover, EVs derived from mesenchymal stem cells have garnered interest in the fields of tissue repair, inflammation, anticancer therapy [19], and stroke [20,21]. Further, compelling evidence marks EVs as a potential drug-delivery system [1,22,23,24]; indeed, engineered EVs are capable of passing through the blood–brain barrier (BBB) [25], which has traditionally been a roadblock for efficient drug delivery to the brain [26,27,28,29].

Besides their beneficial role in the maintenance of physiological homeostasis and potentially therapeutic, diagnostic, and drug-delivery capabilities, EVs have been implicated in many pathogenies, including cardiovascular disease [30], neurodegenerative disorders [31,32,33,34], traumatic brain injury [35,36], HIV [37,38], and a wide range of cancers [39,40,41,42,43]. For instance, EVs may contribute to cancerous proliferation through angiogenesis, migratory and invasive capacities, and formation of metastatic lesions [44]. Dissecting the role and effects of EVs in these disease pathologies presents an ongoing challenge and an opportunity to progress understanding of the mechanisms underlying a diverse array of pressing health issues. Specifically, EV contents may indicate pathological changes in the body, and analysis of the molecular cargoes of the EVs may contribute to the advancement of diagnostic and treatment methods for these diseases.

### 1.2. Extracellular Vesicles in CNS Disorders and Addiction

#### 1.2.1. EVs and CNS Disorders

Central nervous system (CNS) cells like neurons, microglia, astrocytes, oligodendrocytes, ependymal, and brain endothelial cells communicate by releasing EVs containing signaling molecules [45,46]. EVs aid in the signal transmission between neurons and glial cells, along with communication between CNS and peripheral body systems [47,48,49]. EVs maintain cellular homeostasis and clear abnormal aggregates; however, they also contribute to pathogenesis by delivering toxic substances to healthy cells, leading to inflammation and neurodegeneration [50] and thereby perpetuating CNS-associated neurodegenerative disorders [51,52]. Such CNS disorders include lysosomal storage disorders, Parkinson’s disease (PD) [53], Alzheimer’s disease (AD) [54,55,56,57], Huntington’s disease, amyotrophic lateral sclerosis [58], epilepsy, and multiple sclerosis [59,60,61,62,63]. EVs exacerbate disease pathogenesis by providing transportation to abnormally folded proteins and disease factors like α-synuclein [64], amyloid beta (Aβ) and Tau [65,66], huntingtin, and superoxide dismutase 1 [52,58].

EVs in diseased states differ significantly in their morphology and function, making them ideal biomarker candidates [67] as they contain unique proteins depending on the healthy or diseased microenvironment conditions [68,69]. The ability of EVs to cross the BBB, combined with their prevalence in bodily fluids, makes it possible to detect certain biomarkers found in difficult-to-assess regions like the CNS and spleen [70]. EVs may also contribute to neuroprotection; in AD, EVs sequester Aβ in vitro and promote its clearance, thus reducing neurotoxicity [71,72,73]. Moreover, neuronal EVs carry extracellular RNAs [74,75], including disease miRNA signatures that could be used as biomarkers to diagnose CNS disorders [58,76,77,78].

Additionally, EVs are potential candidates as therapeutic delivery agents as they can be easily loaded with therapeutic drugs, are minimally degraded, maintain their morphology and function, and can cross the BBB [2,79,80,81,82]. Due to their ability to carry functional small miRNA, tRNAs, lipids, and proteins [83], EVs are excellent carriers of the therapeutic agents. Besides acting as protective barriers against degradation and immunoreactivity, EVs can also increase the efficiency of delivery to targets, further aiding drug delivery and therapy for CNS diseases.

#### 1.2.2. EVs and Substance Abuse

Investigations into the role of EVs in drug addiction and as future therapeutics for addiction are currently represented by a small but developing body of work [84]. Recent evidence points to a role of EV cargoes, specifically noncoding regulatory miRNAs [85], in mediating the body’s response to a variety of addictive substances, including cocaine [86,87], cannabinoids [88], nicotine [89], alcohol [90], and opioids [91,92]. These studies indicate that EVs and their cargoes may play a significant role in modulating addiction to a variety of substances, but further investigation is required to understand the full impact of EVs on addictive pathways and of addictive substances on EV secretion, uptake, and cargo content. There is a significant gap in the knowledge connecting substance abuse and our understanding of EVs and their cargoes in those addiction pathologies, although many investigators are currently working to close that gap. The present study sought to review the literature investigating the role of EVs in addiction and comorbidities, specifically HIV neuropathology, and briefly highlight the potential of EVs as therapeutics for these pathologies.

## 2. Drugs of Abuse

### 2.1. Stimulants

Stimulants increase stress and hyperactivity of the neural circuitry, which contributes to the brain’s susceptibility to senescence, damage, and dysregulated plasticity [93,94]. Many of the mechanisms impacted by stimulants may be EV-mediated, such as the activation of the inflammatory pathways that may be part of the contiguous cycle of injury associated with stimulant drug abuse and many brain diseases [95,96,97]. The research field of EVs in drug abuse is rapidly growing with regard to the mechanistic understanding of the many comorbidities associated with stimulant abuse, particularly methamphetamine (METH) and cocaine use.

#### 2.1.1. Methamphetamine

METH use disorder is commonly concurrent with increased EV release [98]. The cargoes of these EV remains to be characterized, but many regulatory elements associated with EVs have been implicated in METH abuse and are concomitant with neurodegenerative diseases such as AD and PD [99,100,101]. The vast majority of research into METH abuse and vesicular bodies lies in understanding synaptic vesicles, which are crucial for interneuronal communication, but EVs also contribute to the intercellular communication of the whole nervous system [102]. One paper sequenced the miRNAs found in serum-derived EVs of METH-dependent rats, identifying a total of 301 differentially expressed miRNAs [103]. Four of the differentially regulated miRNA were miR-218b, miR-194-5p, miR-152-3p, and miR-22-3p, which were also noted to be differentially regulated in ketamine dependence [103]. Elevated serum levels of miR-194-5p have been associated with cancers of peripheral organs such as the kidneys and liver [104,105]. Interestingly, elevated miR-194-5p in the serum is also associated with promotion of glioma development as well as generalized epilepsy [106,107]. Both miR-22-3p and miR-152-3p have been shown to be differentially expressed in plasma-derived EV miRNA from AD patients [108]. Of these miRNAs, miR-22-3p might be one of the most promising EV-derived miRNAs concomitant with METH abuse and psychiatric diseases as it has been associated with a wide range of disorders including chronic fatigue syndrome, schizophrenia, bipolar disorder, and attention deficit disorder [109,110,111,112]. Most of these published works have emphasized the differential expression patterns of EV-associated miRNAs, but not much work has been done to elicit mechanisms or pathways to elucidate how these miRNAs contribute to drug seeking, withdrawal, and relapse behaviors. Future work should aim to identify the specific roles of EVs in addiction and related processes.

#### 2.1.2. Cocaine

Closely related to METH in terms of prevalence of use and mechanism of action, cocaine is one of the most prolifically abused stimulants. Recently, cocaine has been shown to stimulate EV release through the sigma-1 receptor (Sig-1R)-ARF6 (ADP-ribosylation factor 6) complex [113]. Further, Nakamura et al. showed that the interactions among Sig1-Rs, cocaine, and EVs may regulate synaptic transmission in the brain through the release of 2-AG (2-arachidonoylglycerol; an endocannabinoid that is increasingly synthesized with cocaine stimulation); this release of 2-AG contributes to the inhibition of GABAergic input to dopamine neurons [113]. Interestingly, in a glioblastoma culture model, cocaine exposure not only increased EV release but also increased tunneling nanotubule (TNT) formation [114,115,116]; both EVs and TNTs are highly correlated with the development of many diseases, such as glioblastoma and neurodegenerative diseases [117,118]. Further, cocaine self-administration has been shown to reduce the internalization of neuronal exosomes, particularly in astrocytes in the nucleus accumbens (NAc); this reduction was then reversed by extinction training [119]. Furthermore, cocaine self-administration alone decreased glial fibrillary acidic protein (GFAP) expression in astrocytes and increased Iba1 expression in microglia. Interestingly, extinction training reversed the increased Iba1 expression in microglia but only partially reversed the reduction of GFAP in astrocytes. GFAP is critical to astrocyte-mediated regulation of axon mylenation and BBB integrity [120], and its reduction has been reported in Down’s syndrome, schizophrenia, bipolar disorder, and depression [121,122]. Further, decreases in GFAP have also been reported in cases of chronic infection with viruses, including HIV [123].

#### 2.1.3. Nicotine

Recently, nicotine has been linked to a multitude of signaling and genetic changes that can be observed molecularly and behaviorally through EV analysis and transcriptomics [124,125,126]. Nicotine acts on nicotinic acetylcholine receptors (nAChRs), ultimately causing an increase in dopamine release and fueling signaling cascades along the reward pathway. Studies implicate nAChRs and miRNAs in aggravating the effects of nicotine [89,127,128]. Elucidation of these specific mechanisms could advance our understanding of EVs in the development of nicotine addiction and subsequent CNS disorders.

Much like the METH and cocaine EV studies, nicotine has been shown to increase the release of EVs [129]. A study of smokers and non-small-cell lung cancer (NSCLC) patients found that over 90% of lung EVs were 50–200 nm in size; additionally, 21 EV miRNAs were upregulated and 10 miRNAs were downregulated in smokers compared to controls [129]. These miRNA were further dysregulated in NSCLC patients compared to smokers. Additionally, this study identified upregulated mRNA transcripts including EGFR, KRAS, ALK, MET, LKB1, BRAF, PIK3CA, RET, and ROS1 in lung EVs in smokers and NSCLC patients. Long noncoding RNAs (lncRNA), including MALAT1, HOTAIR, HOTTIP, AGAP2-AS1, ATB, TCF7, FOXD2-AS1, HOXA11-AS, PCAF1, and BCAR4, also had higher expression levels in EVs from smokers and NSCLC patients. Further, protein levels of tumor-associated antigens, including BAGE, PD-L1, MAGE-3, and AKAP4, were also significantly dysregulated in EVs of smokers and NSCLC. This study concluded that an intrinsic relationship exists between smoking and dysregulated EV secretion and cargo, the contents of which may contribute to the development of NSCLC [129].

With the increasing use of electronic cigarettes (e-cigarettes), it is important to understand how this form of smoking may affect EVs and their cargoes. A recent study found that platelet- and endothelial-derived EVs were increased 4 h after active inhalation of e-cigarette vapor with nicotine [130]. Further, platelet-derived EVs expressing P-selectin, a platelet activation marker, and CD40, an inflammation marker, were significantly increased following inhalation of e-cigarette vapor with nicotine. Interestingly, CD40 expression on plate-derived EVs was also increased by e-cigarette vapor that did not contain nicotine. The study concluded that as few as 30 puffs from a nicotine-containing e-cigarette caused an increase in circulating EVs that originated from endothelial or platelet cells, and nicotine, as a component of the vapor, affects EV formation and protein composition [130].

Nicotine use has recently been shown to have a sex-specific effect pattern on brain-derived EVs (BDEVs) [131]. In a rat self-administration paradigm of nicotine, females had larger BDEV sizes and impaired EV biogenesis compared to males following nicotine self-administration. Using quantitative mass spectrometry to identify changes in BDEV proteins, 2165 and 2051 proteins were found in males and females, respectively. Of these, 10 proteins were upregulated and 21 proteins were downregulated in females. In males, 6 proteins were upregulated and 79 were downregulated. Overall, this study found sex-specific alterations in BDEV biogenesis and cargo content following nicotine self-administration [131].

In the context of disease, nicotine has been found to induce atherosclerotic lesion progression, potentially via EVs. EVs from nicotine-treated macrophages increased proliferation and migration of vascular smooth muscle cells in vitro [132]. After characterizing the miRNA cargo, the researchers found that miR-21-3p was enriched in these EVs. The authors suggested that EV miR-21-3p from nicotine-treated macrophages may accelerate the development of atherosclerosis by increasing VSMC migration and proliferation through PTEN (phosphatase and tension homologue), its target. Further research into nicotine use and EVs is needed to understand the mechanisms and downstream effects that may contribute to subsequent diseases.

### 2.2. Opioids

The increased abuse of prescription opioids, including morphine, oxycodone, fentanyl, and the nonprescription opiate heroin has resulted in a severe public health crisis across large swaths of America [133,134,135]. In 2017, over two-thirds of drug-overdose deaths resulted from opioid abuse [136], and opioid-overdose-attributed deaths have tripled since the turn of the new millennium [137]. This growing public health problem has garnered attention from various scientific communities and has provided heightened motivation to understand addiction pathways and potential therapies to combat opioid addiction, possibly through the use of EVs.

#### 2.2.1. Morphine

Morphine is primarily used as a pain reliever both in the hospital setting and over the counter. While morphine may be effective for pain relief, there are potential adverse side effects such as tolerance and addiction, as well as molecular alterations. In a recent study, the impact of morphine on microglial phagocytosis in the CNS was investigated using a mouse model [138]. This study explored how morphine treatment induces EV release from astrocytes for later uptake by microglia. The astrocyte-derived EVs (ADEVs) taken up by microglia were able to reach the endosomes and activate Toll-like receptor (TLR)7 and TLR8, leading to downstream activation of the nuclear factor κB (NF-κB) signaling pathway. Nearly 1079 of the 2049 identified mouse miRNA hits contained the AU- or GU-rich sequences needed to activate the NF-κB signaling pathway; of those 1079 mouse miRNAs, 38 were found to be present in ADEVs. Of these, 15 were upregulated and 9 were downregulated in morphine-treated ADEVs. These data indicate that morphine-treated ADEV miRNAs could serve as TLR7 and TLR8 agonists in recipient microglia.

Additionally, this study investigated long intergenic noncoding RNA (lincRNA)-Cox2, which is regulated by the NF-κB signaling pathway in microglia [138]. Investigators demonstrated that uptake of morphine-treated ADEVs by microglia upregulated lincRNA-Cox2 via the activation of the TLR7–NK-κB signaling axis, leading to impaired microglial phagocytosis. Morphine-treated ADEVs in mouse primary microglia were also found to decrease expression of several phagocytic genes including *Lrp1*, *Syk,* and *Pld2* [138]. Additionally, the study identified a nearly two-fold increase in ADEV secretion following exposure to morphine, though no differences in EV size distribution between treated and control cells was recorded. While these findings shed light on the role of EVs in microglial function following opioid exposure, this study also demonstrated a lucrative future course for therapeutics in opioid-related neurodegenerative disorders; when delivered intranasally, lincRNA-Cox2 siRNA-loaded ADEVs knocked down lincRNA-Cox2 in morphine-exposed mice microglia and restored microglial phagocytic activity [138].

Similarly, an earlier study examined the potential of EV-delivered siRNA as a therapeutic for opioid addiction [139]. Liu and colleagues utilized rabies viral glycoprotein (RVG) peptides on the membrane surface of EVs to ensure passage through the BBB and receipt by neurons. These RVG EVs contained mu-opioid receptor (MOR) siRNA, which resulted in a reduction in the expression of MORs in target recipient cells. In target cells, levels of MOR mRNA and protein were decreased following entry of RVG EVs. In a behavioral experiment, mice treated with RVG EVs containing MOR siRNA demonstrated restrained drug addiction compared to saline controls following a morphine-administered relapse. These findings further highlight the potential of cargo delivery via EVs for treatment of opioid use disorder (OUD).

#### 2.2.2. Oxycodone

As the opioid epidemic has resulted in a rise in the number of women who present with OUD while pregnant, it is necessary to understand the effects of opioid exposure on fetal development. Interestingly, EVs have recently been investigated in conjunction with fetal development and maternal OUD. A recent study investigated the effects of in utero (IUO) and postnatal (PNO) oxycodone exposure on neurodevelopment via miRNA expression in BDEVs [140]. Using RNA sequencing and bioinformatics, the investigators demonstrated that there was a significant alteration in BDEV miRNA cargo in IUO rats, leading to increased impairment in brain development. Results also indicated synaptodendritic damage to primary neurons following administration of IUO and PNO BDEVs, with enhanced damage to IUO BDEV-exposed neurons. This study provides a number of EV-delivered miRNA signatures associated with development in oxycodone-exposed offspring, which may prove useful in future identification of genes implicated in perinatal development of opioid-exposed offspring. Interestingly, the study also found a significant increase in the size of BDEVs in IUO and PNO rats compared to saline-exposed BDEV controls, but no difference in the number of BDEVs secreted.

#### 2.2.3. Buprenorphine and Methadone

Another recent study investigated the effects of maternal use of methadone and buprenorphine, both commonly used to aid in management of opioid-related withdrawals, on fetal development and EVs [141]. Goetzl and colleagues utilized a novel mechanism designed to isolate fetal neuronal EVs from maternal blood to investigate the effects of maternal OUD on developing fetal brains in humans [142]. Analysis of fetal neuronal EVs isolated from maternal blood revealed increased MOR protein levels in both methadone- and buprenorphine-exposed groups. Of additional interest was the result that cannabinoid receptor expression levels in the EVs were also higher, suggesting crosstalk between cannabinoid and opioid receptors. The study also identified altered EV protein cargo in opioid-exposed subjects compared to controls, although a small sample size limited statistical analysis; further investigation is required to make any concrete conclusions.

### 2.3. Alcohol

Mounting evidence suggests that exposure to alcohol can alter the miRNA, protein, and mRNA content of EVs, as well as altering the proliferation and biogenesis of the EVs themselves [143]. Indeed, a human study of alcohol users with liver injury showed an increase in the total number of EVs as well as increased expression of miR-122 and let7f in blood EVs [144]. Additionally, Ibáñez et al. found that ADEVs from alcohol-treated cell culture contained an increase of inflammatory signaling proteins (TLR4, NFκB-p65, IL-1R, caspase-1) as well as differential expression of miR-146a, miR-182, and miR-200b [145]. Much like METH, cocaine, and nicotine, alcohol also increased the number of EVs. TLR4 knockout cells displayed no change in their EV content compared to untreated cells. Interestingly, neurons that internalized the alcohol-treated ADEVs displayed differential expression of COX-2, Mapk14, IL-1β, Foxo3, Traf6, and miR-146a, all of which are related to inflammation and apoptosis. Indeed, neurons exposed to alcohol-treated ADEVS had a higher rate of apoptosis. Further, TLR4-knockout ADEVs had no effect on the neurons, suggesting that TLR4 is a critical molecule in the inflammation response to alcohol exposure.

While several in vivo studies have suggested that alcohol may regulate the expression of specific miRNA, in vitro work also showed an increased expression of TLR7 and miR-let-7b in response to alcohol treatment [146]. The researchers also found that alcohol facilitated the release of both let-7b and miR-155 in microglial EVs and increased the binding affinity of microvesicular let-7b to HMGB1, contributing to further inflammation. The finding that miR-155 is increased in microglial EVs is consistent with another study which found that alcoholic and inflammatory liver injury led to an increase of microvesicular miR-155 in the plasma [147]. Additionally, miR-155 deficiency or TLR4 knockout protected mice from alcohol-induced neuroinflammation [148]. A fetal neural-stem-cell (NSCs) study found that 47 miRNAs, including miR-140-3p, miR-15b-3p, miR-340-5p, and miR-674-5p, were significantly upregulated in EVs from alcohol-treated NSCs [149]. Overexpression of miR-140-3p increased the number of S-phase cells and decreased the number of G_0_/G_1_ cells, suggesting an increase in cell proliferation. During NSC differentiation, overexpression of miR-140-3p increased the mRNA expression of GFAP (astrocytic marker) and decreased the expression of PDGFR*α* (an oligodendrocyte marker) as well as DCX and NeuN (neuronal markers), potentially promoting aberrant astrocytic differentiation of NSCs at the expense of differentiating to other cell lineages. This dysregulation of miRNA content may contribute to abnormal neurodevelopment linked with fetal alcohol syndrome disorder [149].

EV delivery of miRNA may be critical in the development of alcoholism. An in vitro study of striatal neurons found that miR-9 expression was enhanced, possibly contributing to the development of alcohol tolerance. Scientists found that alcohol treatment of neuronal cells resulted in an increase of miR-9 expression in these cells and stimulated expression alterations in calcium and voltage-gated potassium channels, potentially supporting the development of tolerance for alcohol [84]. Further, intranasal delivery of EVs derived from activated human mesenchymal stem cells (hMSCs) impeded chronic alcohol ingestion and relapse and led to an increase in glutamate transporter expression in rats, counteracting the inhibition of glutamate transporter activity and representing a possible mechanism of inhibition of alcohol intake by EVs [150].

In summary, the current literature on the role of EVs in addiction largely suggests that differential expression of miRNA cargoes contributes to the detrimental effects reported in addictive states. A number of the miRNAs covered in this review have roles in signaling and neurodevelopment. Additionally, dysregulation of several of these miRNAs may contribute to the formation of cancers, lesions, and other CNS disorders. Substances of abuse appear to alter EV cargoes related to inflammation, often resulting in the exacerbation of neuroinflammatory states that subsequently lead to neuropathological issues. The increase in release of EVs following drug exposure may also be critical for the progression of addiction, as these EVs may contain the very cargoes that contribute to the detrimental effects of addiction.

## 3. EVs, Substance Abuse, and HIV

The Centers for Disease Control and Prevention (CDC) reports that out of 38,739 HIV infected individuals in the United States, 9% (3641) are individuals who inject drugs (https://www.cdc.gov/hiv/group/hiv-idu.html). As EVs can cross the BBB, the presence of HIV components in EVs can contribute to neuroinflammation [151] and neurodegeneration [6]. The interactions of HIV and drugs of abuse are of growing interest given the growing incidence of HIV transmission via shared needles during illicit drug use [152,153,154]. HIV exposure may also perpetuate addiction to stimulants [155]. Studies of HIV suggest that neuropathologies and substance abuse disorders often have a complex relationship that cannot be classified in one direction [156,157,158]; HIV and substance use together frequently result in the exacerbation of CNS disorders [159]. EVs are likely a key communication factor causing this exacerbation and interrelationship between HIV and substance abuse [160], however further research needs to be performed.

HIV is particularly hard to treat due to its ability to amass beyond the blood–brain barrier; it has a wide variety of impacts on the brain, including increased EV release [161,162]. Recently, research has investigated the role EVs play in the progression of microglia-mediated inflammation of HIV-infected subjects [151,161,163]. This inflammatory state is not resolved by combination antiretroviral therapy (cART) and remains a persisting issue [151]. Currently, METH is being investigated for its potential role in exacerbating HIV-mediated inflammation due to its ability to increase vesicular shedding and extracellular release [98,159,164,165]. Additionally, macrophage-derived EVs from primary human pulmonary arterial smooth muscle cells have been shown to be critically regulated by cocaine addiction and HIV infection [166].

Much like cocaine and METH, nicotine exacerbates HIV pathogenesis through the oxidative stress pathway [152,167]. Interestingly, EVs have revealed a strong correlation between cigarette smoking and HIV [167]. A recent study found that cigarette smoke condensate (CSC) reduced the total protein and antioxidant capacity in EVs isolated from HIV-infected and uninfected macrophages [168]. The EVs isolated from CSC-treated uninfected cells exhibited a protective property against cytotoxicity and viral replication in HIV-infected macrophages. Intriguingly, EVs isolated from HIV-infected cells lost their protective capacity. Further, levels of catalase and PRDX6, antioxidant enzyme cargoes, were decreased in EVs derived from HIV-infected cells. These results highlight the role of antioxidant enzymes in HIV replication and how the differential packaging of these cargoes into EVs affects nicotine-mediated HIV pathogenesis [168]. Indeed, Ranjit et al. suggest that because neurons have a weak antioxidant defense capacity and therefore rely on astrocytes to supply antioxidants, synthetically developed EVs loaded with antioxidant cargoes may be an efficient strategy for offsetting smoking-induced oxidative stress and HIV replication in the CNS [169].

Previous studies suggest that opioids may also play a role in exacerbating HIV-related neurological dysfunction and neuropathogenesis [170]. In simian immunodeficiency virus (SIV)-infected macaque monkeys, a model of HIV, opioid dependency has been demonstrated to increase mortality and exacerbate viral replication [171]. A 2012 study built upon previous studies of the consequences of HIV infection and opioid use by investigating the role of EV-delivered miR-29b in the regulation of PDGF-B gene expression in opioid-dependent SIV-infected macaques [172]. PDGF-B plays a crucial role in neuronal homeostasis, primarily via the protection of hippocampal neurons from glutamate-induced damage. The results of this study indicated that morphine exposure led to increased miR-29b secretion from astrocytes via EVs and demonstrated that increased miR-29b presentation decreased cell viability via decreased PDGF-B expression. This early study was the first to demonstrate that ADEVs can deliver miRNA cargoes to neurons and, in turn, these cargoes can induce functional changes in gene expression in the recipient neurons.

Similarly, a 2019 study investigated the effects of HIV infection and heroin use on inflammation-associated EV miRNA [173]. This study found that HIV-infected heroin users had significantly upregulated levels of miR-146a, miR-126, miR-21, and miR-let-7a, all of which are implicated in neuroinflammation. Interestingly, only the HIV-infected heroin-using group displayed this upregulation; neither uninfected heroin users nor heroin-free HIV-infected patients displayed significant levels of these miRNAs. Further, several members of the let-7 family of miRNA were significantly upregulated within the group of heroin users without HIV infection, namely miRNA-let-7a, -7d, -7e, -7f, -7g, and -7i. The let-7 family is highly conserved across animal species, including humans and mice, and is known to promote cell differentiation [174]. Interestingly, another group noted that morphine significantly increased expression levels of miRNA-let-7a, 7c, and 7g [91]. These results further indicate the importance of understanding the implications of the combination of HIV infection and opioid use as it relates to EV miRNA cargo.

As opioids and needle-sharing are associated with increased risk of HIV infection, alcohol also increases the risk of infection and aggravates HIV replication. Further, alcohol diminishes the adherence to and the efficiency of antiretroviral therapy (ART), which may further enhance HIV replication. HIV infection is correlated with enhanced expression of pro-inflammatory cytokines and chemokines, consequently promoting the pathogenesis of HIV [175]. In the search for a prospective biomarker for alcohol-stimulated toxicity in HIV patients, Kodidela et al. found that HIV-positive alcohol users had substantially lower levels of EV IL-1ra compared to HIV-negative alcohol drinkers. Additionally, no change in the levels of EV IL-1ra was found in the nondrinker HIV-positive subjects. IL-10 was also present in EVs of HIV-positive drinkers. Furthermore, compared to plasma, the percentages of TNF-α, IL-8, and IL-1ra packaged in the EVs isolated from HIV-positive alcohol users were 15%, 10%, and 10%, respectively [175].

In addition to cytokine EV cargo changes, hemopexin (HPX), a protein that binds to free heme, was found in reduced concentrations in the EVs of HIV-positive drinkers, possibly aggravating or contributing to neuroAIDS in those patients [176]. Although unchanged in alcohol drinkers and HIV patients, HPX was substantially downregulated in alcohol users with HIV. HPX may possess an anti-inflammatory function through the negative regulation of TNF-α and IL-6 secretion by macrophages. Additionally, HPX is an extracellular antioxidant, and its diminished level in the EVs of HIV-positive drinkers is consistent with its protective role against alcohol-induced oxidative stress. Additionally, Kodidela et al. found that GFAP expression was significantly enhanced in plasma EVs obtained from HIV-positive subjects and alcohol users, suggesting increased astrocyte activation in those subjects [177]. Exploring EV cargo alterations, such as those listed in Table 1, may allow the field to progress towards diagnosis of and remedies for alcohol-induced toxicity in HIV patients.

## 4. EVs as Potential Therapeutics for Substance Abuse and HIV-Related Neuropathologies

As EVs have been implicated in the pathophysiology of drug addiction [84], studies have recently reported on the potential of EVs as therapeutic agents [49,180,181]. The biocompatibility, targeting capacity, low immunogenicity, and low toxicity of EVs make EVs attractive candidates for therapeutic delivery systems [1]. Indeed, the administration of EVs, particularly those isolated from stem cells [182], has been shown to ameliorate deleterious effects in disease states. For example, intranasal delivery of EVs isolated from hMSCs adequately distributed EVs into neurons and microglia in intact and injured forebrains, with injured areas showing higher uptake of EVs [183]. Further, intranasal administration of EVs derived from human tooth stem cells improved motor function in a rat model of Parkinson’s disease [184]. Intriguingly, intranasal delivery of modified EVs has shown promise in mitigating negative effects associated with exposure to substances of abuse. Recently, Chivero et al. showed that intranasal administration of EVs loaded with miR-124, a miRNA involved in microglial quiescence, alleviated cocaine-mediated microglial activation [185]. Similarly, ADEVs loaded with siRNA restored phagocytic activity by knocking down lincRNA-Cox2 in morphine-exposed mouse microglia [138]. Additionally, treatment with EVs modified with RVG peptides and loaded with MOR siRNA resulted in restrained drug addiction following morphine-administered relapse [139]. Together, these studies highlight the viability of EVs as a therapeutic option; the ability of EVs to be targeted by injury-related signals and be loaded with specific cargoes are two important factors that may be useful in the treatment of neurological pathologies, including addiction and HIV.

## 5. Conclusions and Future Perspectives

In recent studies, EVs, key players in cell–cell communication throughout the body, have emerged as biological components particularly important for their potential roles in physiological homeostasis, drug delivery systems, and therapeutics. In addition to these roles, EVs are implicated in many pathogenies, including cardiovascular disease [30], neurodegenerative disorders [31,32,33,34], traumatic brain injury [35,36], HIV [37,38], and a wide range of cancers [39,40,41,42,43]. More recently, studies indicate that EVs and their cargoes may play a significant role in modulating addiction across a variety of substances. Indeed, several works have sought to elucidate the role of EVs in addiction and CNS disorders, specifically HIV.

Recent evidence points to the role of EV cargo, specifically noncoding regulatory miRNAs [85], in mediating the body’s response to a variety of addictive substances, such as nicotine [89], ethanol [90], and opioids [91,92]. For example, stimulant use is commonly concurrent with increased EV release as well as changes in EV cargo. Similarly, opioid studies using morphine, oxycodone, buprenorphine, and methadone have all shown unique cargoes of EVs isolated from opioid exposure groups. Further, an increasing number of studies suggest that exposure to alcohol can not only alter the miRNA, protein, and mRNA content of EVs, but also alter the proliferation and biogenesis of the EVs themselves [143]. Additionally, dysregulation of several of these miRNAs may contribute to cancers, lesions, and other CNS disorders. Importantly, drugs of abuse alter EV cargoes related to inflammation, resulting in the exacerbation of neuroinflammatory states that further lead to neuropathological issues. Together, these studies strongly suggest further exploration into the role of EVs in substance use disorders and provide solid groundwork for future investigations.

While EVs have been shown to play a role in CNS disorders, the intersection of EVs, drug use, and HIV is of particular interest. The interactions of HIV and drugs of abuse are a growing concern given the increasing incidence of HIV transmission via shared needles in illicit drug use. As a drug commonly taken through shared needles, METH is being investigated due to its role in exacerbating HIV-mediated inflammation through both increased vesicular shedding and extracellular release. In vivo experiments have shown that cocaine-induced EV release impacts synaptic plasticity through noncoding RNA. Nicotine studies have also highlighted how the differential packaging of antioxidant enzyme cargoes into EVs affects nicotine-mediated HIV pathogenesis. Additionally, studies of both morphine and heroin have demonstrated differences in the miRNA cargoes of EVs, potentially impacting gene expression and exacerbating HIV. Studies of alcohol use in combination with HIV have shown that EV cargoes such as cytokines are affected in HIV-infected subjects who use alcohol. Investigating EV cargo alterations in all forms of substance abuse studies may allow the EV, HIV, and addiction fields to progress towards diagnosis and remedies for substance-abuse-induced toxicity in HIV patients.

Taken together, the studies presented in this comprehensive review shed light on the potential of EVs to exacerbate substance use and HIV (Graphical Abstract). As a persistently growing subject of interest, EVs may continue to provide information about mechanisms and pathogenesis in substance use disorders and CNS pathologies, perhaps allowing for exploration into potential therapeutics. Future studies should aim to build on the present works and investigate the altered EV cargoes that may be critical for intercellular communication, immune response, antigen presentation, and signal transduction, all of which can contribute to the exacerbation of pathologies when dysfunctional.

## Figures and Tables

**Table 1 ijms-21-06765-t001:** Differentially regulated EV cargoes identified in studies of substance abuse and HIV.

Cargo		Condition	EV Source	Model	Up/Down	Reference
**miRNA**	29b	Morphine + HIV	Astrocyte	Rat primary cultures	Up	[172]
	21	Heroin + HIV	Plasma	Human	Up	[173]
	146a	Heroin + HIV	Plasma	Human	Up	[145,173]
	126	Heroin + HIV	Plasma	Human	Up	[173]
	let-7a	Heroin + HIV	Plasma	Human	Up	[173]
	let-7b	Alcohol	Microglia	BV2 cell line	Up	[146]
	276	Methamphetamine (METH)	Plasma	Rat	Up	[103]
	218b	METH	Plasma	Rat	Up	[103]
	194-5p	METH	Plasma	Rat	Up	[103]
	152-3p	METH	Plasma	Rat	Up	[103]
	25	METH	Plasma	Rat	Down	[103]
	276	Ketamine	Plasma	Rat	Down	[103]
	22-3p	METH/Bipolar	Plasma	Rat	Up	[103,110]
	107	Nicotine	Bronchoalveolar lavage fluid (BLF)	Human	Up	[129]
	126	Nicotine	BLF	Human	Up	[129]
	19a-3p	Nicotine	BLF	Human	Up	[129]
	200a-3p	Nicotine	BLF	Human	Up	[129]
	21-3p	Nicotine	Macrophage	RAW264.7 cell line	Up	[132]
	21	SIV	Brain	Monkey	Up	[160]
	182	Alcohol	Astrocyte	Mouse primary culture	Up	[145]
	200b	Alcohol	Astrocyte	Mouse primary culture	Down	[145]
	155	Alcohol	Microglia	BV2 cell line	Up	[146]
	140-3p	Alcohol	Fetal neural stem cells (fNSC)	Mouse	Up	[149]
	15b-3p	Alcohol	fNSC	Mouse	Up	[149]
	340-5p	Alcohol	fNSC	Mouse	Up	[149]
	674-5p	Alcohol	fNSC	Mouse	Up	[149]
	130a	HIV/Cocaine	Monocytes	Monomac-1 cell line	Up	[166]
**lncRNA**	MALAT1	Nicotine	BLF	Human	Up	[129]
	HOTAIR	Nicotine	BLF	Human	Up	[129]
	HOTTIP	Nicotine	BLF	Human	Up	[129]
	AGAP-AS1	Nicotine	BLF	Human	Up	[129]
	ATB	Nicotine	BLF	Human	Up	[129]
	TCF7	Nicotine	BLF	Human	Up	[129]
	FOXD2-AS1	Nicotine	BLF	Human	Up	[129]
	HOXA11-AS	Nicotine	BLF	Human	Up	[129]
	PCAF1	Nicotine	BLF	Human	Up	[129]
	BCAR4	Nicotine	BLF	Human	Up	[129]
**mRNA**	EGFR	Nicotine	BLF	Human	Up	[129]
	KRAS	Nicotine	BLF	Human	Up	[129]
	ALK	Nicotine	BLF	Human	Up	[129]
	MET	Nicotine	BLF	Human	Up	[129]
	LKB1	Nicotine	BLF	Human	Up	[129]
	BRAF	Nicotine	BLF	Human	Up	[129]
	PIK3CA	Nicotine	BLF	Human	Up	[129]
	RET	Nicotine	BLF	Human	Up	[129]
	ROS1	Nicotine	BLF	Human	Up	[129]
**Cytokines**	130a	HIV/Cocaine	Monocytes; Plasma	Monomac-1 cell line; Human	Up	[166,175]
	IL6/IL-8	Smoking + HIV	Plasma	Human	Up	[175]
	IL-6	Smoking + HIV	Plasma	Human	Up	[175]
	IL-1ra	Alcohol/ Nicotine + HIV	Plasma	Human	Up	[175]
	IL-10	Alcohol/Nicotine HIV	Plasma	Human	Up	[175]
**Proteins**	Amyloid beta (Aβ)	HIV	Brain	Human	Up	[161]
	GFAP	HIV + Alcohol	Plasma	Human	Up	[177]
	L1CAM	Nicotine	Plasma	Human	Up	[177]
	α-synuclein	METH	Neuroblastoma cells	SH-SY5Y cell line	Up	[178]
	TLR4	Alcohol	Astrocyte	Mouse primary culture	Up	[145]
	NFκB-p65	Alcohol	Astrocyte	Mouse primary culture	Up	[145]
	IL-1R	Alcohol	Astrocyte	Mouse primary culture	Up	[145]
	Caspase-1	Alcohol	Astrocyte	Mouse primary culture	Up	[145]
	CPM	HIV	Plasma	Human	Up	[179]
	CDH3	HIV	Plasma	Human	Up	[179]
	HPX	HIV + alcohol	Plasma	Human	Down	[176]
	BAGE	Nicotine	Lung	Human	Up	[129]
	PD-L1	Nicotine	Lung	Human	Up	[129]
	PRDX6	HIV + Nicotine	Macrophage	U937 cells	Down	[168]
	Catalase	HIV + Nicotine	Macrophage	U937 cells	Down	[168]
	CSF2RA	HIV	Plasma	Human	Up	[179]
	MANF	HIV	Plasma	Human	Up	[179]

Abbreviations: METH: Methamphetamine; BLF: bronchoalveolar lavage fluid; fNSC: fetal neural stem cells.

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
