# Peer review of "Role of Extracellular Vesicles in Substance Abuse and HIV-Related Neurological Pathologies"

_ijms, 2020, doi:10.3390/ijms21186765_

Round 1
Reviewer 1 Report
The review titled “ Role of Extracellular Vesicles in substance abuse and neurological pathologies” summarized the recent studies of EVs in the context of METH, cocaine, nicotine, opioid, and alcohol, revealed the important role of EV cargos in substance abuse-related neurological pathologies, and then discussed the potential contribution of EVs in the pathogenesis of substance use and HIV, which indicates that the detail studies in EVs provide potential therapeutics in substance abuse mediated CNS pathologies. Overall, the review is well written and easy to grasp. I would suggest the authors enrich some sections which are a little bit short on information and necessary details. For me, it will be a minor revision. The comments as below:
- There are several places that missed the citation—for example, page 1, line 33; page 3, line 116. On page 5, line 219, the author mentioned ‘another study ‘ but has the same citation as the previous paragraph.
- Spell checking, page 4, line 158, ‘further our understanding’, the verb is missing. Grammar checking is also needed.
- Page 4, line 146, the author gave an example that cocaine self-administration was showed to drastically increase EV signaling to astrocytes and microglia in the NAc, which leading to a decrease in GFAP. Later, the author commented that GFAP increase is considered a hallmark of neurodegenerative disorders that are not well explained. More information related to these studies is needed.
- The author conducted a schematic figure showing the involvement of EVs released from various types of cells in drug abuse or HIV infection mediated addiction and neurological disorders. However, if the authors could summarize the identified critical EVs cargos related to substance abuse in a table, it will bring more information and be more accessible.
- The author highlighted the importance of miR-22-3p in both cocaine and METH sections. More discussions about the miR-22-3p should be included, for example, does this mean that cocaine and METH has a similar effect. If it is, what will it be? If not, what’s the difference?
- As the author mentioned that “EVs may continue to provide information about mechanisms and pathogenesis in substance use disorders and CNS pathologies, perhaps allowing for exploration into potential therapeutics.” It will be interesting that the author adds a section that discusses the development of a therapeutic strategy based on the EVs. For example, studies used intranasal delivery of EVs to ameliorate the substance abuse-related neuroinflammation or CNS pathologies.
Author Response
We sincerely thank the reviewers for their helpful criticisms. We have modified the manuscript accordingly. Our responses are bullet-pointed and italicized below the reviewers comments.
Reviewer 1
The review titled “Role of Extracellular Vesicles in substance abuse and neurological pathologies” summarized the recent studies of EVs in the context of METH, cocaine, nicotine, opioid, and alcohol, revealed the important role of EV cargos in substance abuse-related neurological pathologies, and then discussed the potential contribution of EVs in the pathogenesis of substance use and HIV, which indicates that the detail studies in EVs provide potential therapeutics in substance abuse mediated CNS pathologies. Overall, the review is well written and easy to grasp. I would suggest the authors enrich some sections which are a little bit short on information and necessary details. For me, it will be a minor revision. The comments as below:
- There are several places that missed the citation—for example, page 1, line 33; page 3, line 116. On page 5, line 219, the author mentioned ‘another study ‘ but has the same citation as the previous paragraph.
- The authors thank the reviewer for this comment. Sources have been added for Page 1, Line 33 and for Page 3, Line 116. The citation issue on Page 5, line 219, mentioned by the review has been corrected. The line numbers are now Page 5, lines 219-220.
- Spell checking, page 4, line 158, ‘further our understanding’, the verb is missing. Grammar checking is also needed.
- In the context of this sentence, “further” is in its verb form: help the progress or development of (something); promote. In this context, “could” serves as a helping verb in the complete verb phrase “could further.” Respectfully, the authors maintain there is no grammar issue in this sentence. However, we have changed “further” to “advance” in the hope of remedying any confusion.
- Page 4, line 146, the author gave an example that cocaine self-administration was showed to drastically increase EV signaling to astrocytes and microglia in the NAc, which leading to a decrease in GFAP. Later, the author commented that GFAP increase is considered a hallmark of neurodegenerative disorders that are not well explained. More information related to these studies is needed.
- The authors thank the reviewer for this suggestion. The GFAP discussion has been rewritten to focus on GFAP reduction rather than increase, as this is what the preceding sentences regarding a cocaine self-administration study discuss. We hope focusing on GFAP decrease in the context of diseases and disorders rather than GFAP increase provides a better link for discussion concerning GFAP and its reduction with cocaine use as well as other diseases and disorders. This rewritten discussion can be found on Page 4, lines 142-151.
- The author conducted a schematic figure showing the involvement of EVs released from various types of cells in drug abuse or HIV infection mediated addiction and neurological disorders. However, if the authors could summarize the identified critical EVs cargos related to substance abuse in a table, it will bring more information and be more accessible.
- The authors agree that this is an important table to include. We have included this table of differentially regulated EV cargoes on Pages 9-11.
- The author highlighted the importance of miR-22-3p in both cocaine and METH sections. More discussions about the miR-22-3p should be included, for example, does this mean that cocaine and METH has a similar effect. If it is, what will it be? If not, what’s the difference?
- The authors thank the reviewer for bringing this to our attention. Upon revisiting this section, we found a lack of relevant sources and the inclusion of incorrect sources. To remedy this, the Cocaine section has been rewritten to include a different discussion regarding cocaine’s effects on EV release (Page 3, lines 133-139). Discussion regarding miR-22-3p remains in the METH section, as this miRNA was correctly cited and is correct in that context of discussion. Because we did not have any papers regarding miR-22-3p and cocaine use, we are unable to discuss this miRNA’s effects in the context of cocaine and METH. The authors apologize for this mistake.
- As the author mentioned that “EVs may continue to provide information about mechanisms and pathogenesis in substance use disorders and CNS pathologies, perhaps allowing for exploration into potential therapeutics.” It will be interesting that the author adds a section that discusses the development of a therapeutic strategy based on the EVs. For example, studies used intranasal delivery of EVs to ameliorate the substance abuse-related neuroinflammation or CNS pathologies.
- The authors thank the reviewer for this suggestion. A therapeutics section has been included as section 4 (Page 9).

Reviewer 2 Report
This is a comprehensive review on the role of EVs on substance abuse and neurological pathologies that cover all types of substance abuse. The authors have done good job in covering these substance abuse in details and in context to neurological effects of these substance abuse. However, it also covers HIV in context to substance abuse. Therefore HIV should also be included in the title. If not, then its important to justify why only HIV is covered and not other diseases that also affect neuropathologies in context to these substance abuse.
The other minor issue is that a few important references in context to EVs in smoking/nicotine and alcohol and HIV, especially when they deal with the CNS EV factors, e.g. Kodidela et al., Diagnostics, 2020, Haque et al., Cells, 2020, Ranjit et al., Exp Opin in Therap Targets, 2018
Author Response
RESPONSE TO REVIEWER’S COMMENTS
We sincerely thank the reviewers for their helpful criticisms. We have modified the manuscript accordingly. Our responses are bullet-pointed and italicized below the reviewers comments.
Reviewer 2
This is a comprehensive review on the role of EVs on substance abuse and neurological pathologies that cover all types of substance abuse. The authors have done good job in covering these substance abuse in details and in context to neurological effects of these substance abuse. However, it also covers HIV in context to substance abuse. Therefore HIV should also be included in the title. If not, then its important to justify why only HIV is covered and not other diseases that also affect neuropathologies in context to these substance abuse.
- The title has been changed according to this suggestion. The title now reads “Role of Extracellular Vesicles in Substance Abuse and HIV-related Neurological Pathologies.”
The other minor issue is that a few important references in context to EVs in smoking/nicotine and alcohol and HIV, especially when they deal with the CNS EV factors, e.g. Kodidela et al., Diagnostics, 2020, Haque et al., Cells, 2020, Ranjit et al., Exp Opin in Therap Targets, 2018
- The authors thank the reviewer for bringing these pertinent papers to our attention. They have been added to the relevant smoking/nicotine and alcohol and HIV sections.
